# Identification of Clinical Value and Biological Effects of *XIRP2* Mutation in Hepatocellular Carcinoma

**DOI:** 10.3390/biology13080633

**Published:** 2024-08-19

**Authors:** Dahuan Li, Xin Bao, Shan Lei, Wenpeng Cao, Zhirui Zeng, Tengxiang Chen

**Affiliations:** 1Engineering Research Center of Chronic Disease Diagnosis and Treatment, Guizhou Medical University, Guiyang 550025, China; lidahuan@gmc.edu.cn (D.L.); baoxin@stu.gmc.edu.cn (X.B.); leishan@gmc.edu.cn (S.L.); 2Department of Physiology, School of Basic Medicine, Guizhou Medical University, Guiyang 550025, China; 3Department of Anatomy, School of Basic Medicine, Guizhou Medical University, Guiyang 550025, China; caowenpeng@gmc.edu.cn

**Keywords:** hepatocellular carcinoma (HCC), *XIRP2*, mutation, clinical value, biological effects

## Abstract

**Simple Summary:**

Numerous genetic mutations have been documented in HCC, and associated with clinical characters. Herein, we demonstrated that *XIRP2* mutation is one of the high-frequency mutations in HCC, and is associated with poor prognosis and drug resistance. The presence of the *XIRP2* mutation leads to enhanced stability of the XIRP2 protein, whereas its depletion significantly heightens sensitivity to oxaliplatin by inducing an excessive influx of zinc and calcium ions. *XIRP2* mutation holds potential as a prognostic biomarker and that targeting the XIRP2 protein may offer therapeutic benefits in the management of HCC.

**Abstract:**

Hepatocellular carcinoma (HCC) is a prevalent malignant digestive tumor. Numerous genetic mutations have been documented in HCC, yet the clinical significance of these mutations remains largely unexplored. The objective of this study is to ascertain the clinical value and biological effects of xin actin binding repeat containing 2 (*XIRP2*) mutation in HCC. The gene mutation landscape of HCC was examined using data from the Cancer Genome Atlas and the International Cancer Genome Consortium databases. The prognostic significance of the *XIRP2* mutation was assessed through KM plot analysis. The association between drug sensitivity and the *XIRP2* mutation was investigated using the TIDE algorithm and CCK-8 experiments. The biological effects of the *XIRP2* mutation were evaluated through qRT-PCR, protein stability experiments, and relevant biological experiments. The *XIRP2* mutation is one of the high-frequency mutations in HCC, and is associated with poor prognosis. A total of 72 differentially expressed genes (DEGs) were observed in HCC tissues with the *XIRP2* mutation as compared to those with the *XIRP2* wildtype, and these DEGs were closely related to ion metabolic processes. The *XIRP2* mutation was linked to alterations in the sensitivity of fludarabine, oxaliplatin, WEHI-539, and LCL-161. CCK-8 assays demonstrated that HCC cells carrying the *XIRP2* mutation exhibited increased resistance to fludarabine and oxaliplatin, but enhanced sensitivity to WEHI-539 and LCL-161 as compared with those HCC cells with the *XIRP2* wildtype. The *XIRP2* mutation was found to have no impact on the mRNA levels of XIRP2 in tissues and cells, but it did enhance the stability of the XIRP2 protein. Mechanically, the inhibition of *XIRP2* resulted in a significant increase in sensitivity to oxaliplatin through an elevation in zinc ions and a calcium ion overload. In conclusion, the *XIRP2* mutation holds potential as a biomarker for predicting the prognosis and drug sensitivity of HCC and serves as a therapeutic target to enhance the efficacy of oxaliplatin.

## 1. Introduction

Hepatocellular carcinoma (HCC) is a malignancy that arises from hepatocytes and ranks as the fourth most prevalent digestive malignancy globally, exhibiting a significantly high mortality rate [1]. The development of HCC is gradual, and its initial symptoms lack specificity. While the combination of AFP testing and imaging examinations can detect some cases of early-stage HCC, the majority of diagnosed patients are already in advanced stages, resulting in a five-year survival rate of approximately 18%, which is considerably lower than that of patients diagnosed at an early stage [2]. The high rate of metastasis and recurrence are features of HCC. Nevertheless, the effectiveness of clinical interventions varies significantly due to inherent drug resistance [3,4]. Consequently, the identification of novel diagnostic markers and therapeutic targets assumes paramount importance.

Mutation serves as a significant hallmark of cancer, encompassing proliferation, infiltration, metastasis, and drug resistance [5]. Specific genes hold the potential to anticipate the clinical response to certain oncology medications. For instance, non-small-cell lung cancer requires the presence of *EGFR*, *ALK*, *ROS1*, *BRAF*, *MET*, *RET*, and *KRAS* genes [6], while the prognosis of colorectal cancer patients can be determined by the mutation of *KRAS*, *NRAS*, *BRAF*, and *HER2* genes, as well as microsatellite instability analysis [7,8]. Gene mutations can result in conformational changes in corresponding proteins, DNA repair deficiencies, or resistance to oncology drugs [9]. Research has indicated that *RAS* mutations serve as a reliable indicator of tumor resistance to cetuximab or panitumumab [10]. However, there is no specific gene mutation that can serve as a predictive factor for the treatment of HCC.

Xin actin binding repeat containing 2 (XIRP2) is a member of the actin-binding, xin-repeat-containing protein family [11]. Under normal physiological conditions, the XIRP2 protein exhibits high expression levels in skeletal and cardiac muscle cells, where it plays a protective role in preventing actin filament depolymerization [12]. A previous study has demonstrated that the *XIRP2*-mutation is a prevalent mutation (>3%) in gastric cancer, and its presence is associated with a negative prognosis [13]. However, the understanding of the prevalence and impact of *XIRP2* mutation on HCC remains limited.

The current study aims to analyze the prevalence of the *XIRP2* mutation and investigate its correlation with clinical characteristics and biological functions through bioinformatics analysis and experimental investigations. Our study revealed that the *XIRP2* mutation is prevalent in HCC and is strongly correlated with unfavorable prognosis and drug resistance. The presence of the *XIRP2* mutation leads to enhanced stability of the XIRP2 protein, whereas its depletion significantly heightens sensitivity to oxaliplatin by inducing an excessive influx of zinc and calcium ions. Consequently, we propose that the *XIRP2* mutation holds potential as a prognostic biomarker and that targeting the XIRP2 protein may offer therapeutic benefits in the management of HCC.

## 2. Materials and Methods

### 2.1. Acquisition of Somatic Variant Profile and Somatic Variant Analysis

The somatic variant profiles and clinical characteristics of HCC tissues were obtained from the Cancer Genome Atlas (TCGA; https://portal.gdc.cancer.gov, accessed on 7 July 2023) and the International Cancer Genome Consortium (ICGC; https://dcc.icgc.org/, accessed on 7 July 2023). The VarScan algorithm was employed to detect somatic variant profiles in HCC tissues for gene mutation analysis. The top 10 gene mutations in HCC were visualized using the maftools package [14]. Subsequently, the HCC patients were categorized into wildtype and mutation groups, and the impact of gene mutation on HCC prognosis was assessed through the Kaplan survival analysis. The hazard rate (HR) and *p* value of the mutations were shown in a forest plot.

### 2.2. Acquisition of Gene Expression Profile and Differentially Expressed Genes (DEGs) Analysis

The RNA-seq profiles of HCC in the Cancer Genome Atlas (TCGA) and the International Cancer Genome Consortium (ICGC) were obtained. Prior to analysis, the matrices were normalized using the limma package (version: 3.6.4), and the probe names were converted to gene names. The differentially expressed genes (DEGs) between HCC tissues with wildtype and mutated genes were then analyzed in R software (version 4.1.2). DEGs were defined as genes with an absolute value of log fold change ≥1 and an adjusted *p* value < 0.05. The volcano plot displays the log fold change and the adjusted *p* values for all genes, while the heatmap highlights the DEGs.

### 2.3. Biological Process (BP) Enrichment Analysis

The genes were imported into a database, named the Database for Annotation, Visualization, and Integrated Discovery (DAVID; https://david.ncifcrf.gov/, accessed on 16 July 2023), in order to obtain the enriched biological process (BP) terms associated with them. BP terms with a statistical significance of *p <* 0.05 were considered significant and subsequently visualized in a bubble diagram using the R software.

### 2.4. Drug Sensitivity Calculation

The OncoPredict R package was utilized to predict the in vivo drug responses of 198 drugs in patients with HCC [15]. This package facilitated the acquisition of drug scores for each drug by fitting the gene expression matrix of HCC tissues to the gene expression matrix of cancer lines from the Broad Institute Cancer Cell Line Encyclopedia (CCLE; https://sites.broadinstitute.org/ccle/, accessed on 26 July 2023), as well as the corresponding half-maximal inhibitory concentration of drugs to cancer cells. A higher drug score indicates a greater resistance of patients to the drugs. The drug scores between HCC patients with gene mutation and wildtype were analyzed by *t*-test, with the cut-off as *p* < 0.05.

### 2.5. Cell Culture and Transfection of Plasmids and Small Interfering RNA (siRNA)

The HCC cell lines HepG2, Hep3B, HuH7, Huhi, and SK-Hep1 were procured from Procell (Wuhan, China), while the SNU475 cells were obtained from Icellbioscience (Shanghai, China). Based on the mutation information obtained from the CCLE database, the HCC cell line SNU475 exhibited the XIRP2 mutation (I827V), while HepG2, Hep3B, HuH7, HuH1, and SK-Hep1 displayed the XIRP2 wildtype. All HCC cells were cultured in Dulbecco’s modified eagle medium supplemented with 10% fetal bovine serum at a temperature of 37 °C in an environment with 5% CO_2_. The XIRP2-overexpression plasmids were generated by subcloning the PCR-amplified, full-length human XIRP2 cDNA into the pRFP-C-RS vector and were obtained from Genechem (Shanghai, China). The targeting siRNAs for the XIRP2 used for the current study were bought from iGenebio (Beijing, China). The transfection of the plasmids and siRNAs were both conducted using lipo2000 (Thermo Fisher Scientific, Boston, MA, USA).

### 2.6. Western Blotting

The extraction of the total protein from each sample was conducted using a high-intensity RIPA lysis buffer (Sangon, Shanghai, China) supplemented with a 1/100 concentration of phenylmethanesulfonyl fluoride (Sangon, Shanghai, China). The protein concentrations were measured using a bicinchoninic acid detection kit (Sangon, Shanghai, China). The electrophoresis and membrane transfer of the protein samples were carried out following the procedures described in a previous study [16]. Next, the protein-containing membranes underwent incubation with anti-GAPDH (1:50,000; cat no. 60004-1-Ig, Proteintech, Wuhan, China) and anti-XIRP2 antibodies (1:500; cat no. 11896-1-AP, Proteintech, Wuhan, China). Subsequently, the membranes were washed twice with TBST and incubated with secondary antibodies conjugated with horseradish peroxidase. The detection of the protein bands was achieved by employing chemiluminescence reagents (Sangon, Shanghai, China). The expression levels of XIRP2 were then normalized to GAPDH.

### 2.7. Ion Level Detection

The concentrations of zinc ion and calcium ion within the cells were measured using the TSQ probe (cat no. AAT-21254, AAT Bioquet, Boston, MA, USA) and the Fluo-4 AM probe (Beyotime, Nanjing, China). The probes were prepared and applied in accordance with the manufacturers’ instructions, and the detection was performed using spectrophotometry and immunofluorescence techniques. The specific excitation and emission wavelengths for the TSQ and Fluo-4 AM probes were as follows: TSQ, Ex = 344 nm/Em = 385 nm; Fluo-4 AM, Ex = 488 nm/Em = 512 nm.

### 2.8. Cell Count Kit-8 (CCK-8) Assays

HCC cells were seeded in 96-well plates at a density of 4 × 10^3^ cells/well and incubated at 37 °C. Once the cells adhered, varying concentrations of fludarabine (Cat no. HY-B0069, MCE, Wuhan, China), oxaliplatin (Cat no. HY-17371, MCE), WEHI-539 (Cat no. HY-15607, MCE), and LCL-161 (Cat no. HY-15518, MCE) were added to the HCC cells for a duration time. Subsequently, the medium containing the drugs was removed, and each well was supplemented with 100 μL of fresh medium containing 1/10 CCK-8 reagent (Yeasen, Shanghai, China) for a 2-h incubation period. Finally, the absorbency of each well was measured at a multi-scan spectrum (Bio-rad, Sacramento, CA, USA) under a 450 nm wavelength. These experiments were performed thrice.

### 2.9. Sphere-Forming Experiment

In the 3D sphere formation experiment, a total of 500 HCC cells were suspended in a 100 μL culture medium and placed in ultra-low adsorption petri dishes (Invitrogen, Boston, MA, USA). After a period of 20 days, the condition of the spheres that originated from HCC cells was documented.

### 2.10. TUNEL Stain

HCC cells were cultured in a 24-well plate and exposed to a specific treatment for 48 h. Subsequently, the cells were fixed with 4% paraformaldehyde for 30 min and washed with PBS. Following this, a solution of 0.3% Triton-X 100 in PBS was applied and incubated for 5 min. The apoptosis efficiency was assessed using the one-step TUNEL apoptosis detection kit (Beyotime, Shanghai, China) as per the manufacturer’s guidelines.

### 2.11. Statistical Analysis

Data statistics were performed in Prism 6 software. The differences between the groups were analyzed by the one-way analysis of variance combined with Tukey’s post detection. *p* < 0.05 was set as the cut-off to determine significance.

## 3. Results

### 3.1. Aanalysis of Landscape of Somatic Variant in HCC Tissues

Through analysis of the somatic variant profile of HCC tissues in the TCGA database, we identified the top 20 genes with high frequency mutations. These genes include *TP53*, *TTN*, *CTNNB1*, *MUC16*, *ALB*, *PCLO*, *MUC4*, *RYR2*, *ABCA13*, *APOB*, *GSMD3*, *LRP1B*, *FLG*, *OBSCN*, *XIRP2* (7.9%), *AXIN1*, *ARID1A*, *HMCN1*, *CACNA1E*, and *RYR1* (Figure 1A). Furthermore, in HCC tissues from the ICGC, we observed a similar pattern with the top 20 high-frequency mutation genes being *TTN*, *TP53*, *CTNNB1*, *PCLO*, *MUC16*, *APO8*, *ALB*, *LRP1B*, *GSMD3*, *ARID1A*, *XIRP2* (8.1%), *EYS*, *DST*, *ADGRV1*, *MUC19*, *ARID2*, *RYR2*, *ABCA13*, *HMCN1*, and *DNAH7* (Figure 1B). A total of 15 high frequency mutation genes, including *TP53*, *TTN*, *CTNNB1*, *MUC16*, *PCLO*, *APOB*, *ALB*, *LRP1B*, *GSMD3*, *ARID1A*, *XIRP2*, *ADGRV1*, *RYR2*, *ABCA13*, and *HMCN1*, were identified as being overlapped in HCC tissues from both the TCGA and ICGC datasets (Figure 1C). Among these overlapped high-frequency mutation genes, the *XIRP2* mutation was found to be significantly associated with poor prognosis in HCC tissues in both the TCGA database (HR = 1.93, 95% CI = 1.09–3.40; Figure 1D) and the ICGC database (HR = 1.90, 95% CI = 1.20–3.01; Figure 1E). These results indicated that the *XIRP2* mutation can act as a widely applicable biomarker for predicting the prognosis of HCC patients. Therefore, we focused on the *XIRP2* mutation in HCC. In detail, *XIRP2* mutations are the main form of single-nucleotide missense mutations (Figure 1A,B). Furthermore, we found that the single-nucleotide missense mutation of *XIRP2* was highly prevalent in exon 8 of *XIRP2* (Figure 1F). Moreover, we found that more HCC patients with the *XIRP2* mutation had high fibrosis Ishak scores (Figure 1G). By performing a multi-factor regression analysis, we also found that the *XIRP2* mutation can act as an independent factor for HCC patient survival in the TCGA cohort (Figure 1H).

### 3.2. Identification of Molecular Landscape Affected by XIRP2 Mutation in HCC

We subsequently investigated the molecular landscape impacted by the *XIRP2* mutation in HCC. Differential expression analyses were conducted on HCC tissues with the *XIRP2* mutation and the *XIRP2* wildtype using the TCGA and ICGC databases. The findings revealed a total of 685 upregulated genes and 581 downregulated genes (Figure 2A,B) in the TCGA dataset, while the ICGC dataset showed 670 upregulated genes and 738 downregulated genes (Figure 2C,D) in the *XIRP2* mutation as compared to the *XIRP2*-wildtype HCC tissues. Notably, there were 33 upregulated genes (Figure 2E) and 39 downregulated genes (Figure 2F) that exhibited overlapping expression patterns in both the TCGA and ICGC datasets.

Through the implementation of the BP term enrichment analysis, it was determined that the 33 upregulated DEGs exhibited enrichment in the following terms: “regulation of cellular adhesion to growth factor stimulation”, “intercellular adhesion via plasma membrane adhesion molecules”, “cellular response to zinc ions”, “metal ion response to stress”, and “copper ion response to stress” (Figure 2G). By utilizing the BP term enrichment analysis, it was ascertained that the 39 downregulated DEGs displayed enrichment in various terms, including “proton transport across the membrane”, “monovalent inorganic cation homeostasis”, “reactions of purine-containing compounds”, “organophosphorus reactions”, and “transfer metal ion transport” (Figure 2H). Notably, a majority of the BP terms influenced by the DEGs of the *XIRP2* mutation were linked to ion metabolism. Consequently, it can be inferred that the *XIRP2* mutation significantly impacts the process of metal ion metabolism.

### 3.3. XIRP2 Mutation Increased the Resistance of HCC Cells to Fludarabine and Oxaliplatin but Increased the Sensitivity to WEHI-539 and LCL 161

The OncoPredict algorithm was employed to assess the potential impact of the *XIRP2* mutation on drug sensitivity in HCC. A total of 198 drugs were subjected to analysis, revealing significant alterations in the drug score of four drugs in HCC tissues with the *XIRP2* mutation (Figure 3A). Notably, fludarabine (Figure 3B) and oxaliplatin (Figure 3C) exhibited elevated drug scores in HCC tissues with the *XIRP2* mutation, suggesting a potential resistance to these drugs in HCC tissues with the *XIRP2* mutation. Similarly, our findings indicate a decrease in the drug score of WEHI-539 (a Bcl-2 inhibitor; Figure 3D) and LCL 161 (an IAP inhibitor; Figure 3E) in HCC tissues with the *XIRP2* mutation. This suggests that the HCC tissues with the *XIRP2* mutation may exhibit heightened sensitivity to WEHI-539 and LCL 161.

To validate these findings, we employed the CCLE database to collect data on the *XIRP2* mutation in HCC cells. We identified an HCC cell line (SNU475) carrying the *XIRP2* mutation (I827V), as well as five HCC cell lines (HepG2, Hep3B, Huh7, Huh1, and SK-Hep1) with the *XIRP2* wildtype, which were utilized in our experiments. By conducting CCK8 assays, we determined that the IC50s of fludarabine at 48 h for SNU475, HepG2, Hep3B, Huh7, Huh1, and SK-Hep1 were 6.24 μM, 2.13 μM, 2.53 μM, 2.90 μM, 3.82 μM, and 3.66 μM, respectively (Figure 3F). The IC50 values for oxaliplatin were found to be 16.34 μM, 6.78 μM, 7.53 μM, 4.12 μM, 6.24 μM, and 4.86 μM in SNU475, HepG2, Hep3B, Huh7, Huh1, and SK-Hep1, respectively (Figure 3G). Similarly, the IC50 values for WEHI-539 were determined to be 0.84 μM, 2.95 μM, 3.93 μM, 4.98 μM, 4.24 μM, and 4.73 μM for the same cell lines (Figure 3H). Lastly, the IC50 values for LCL 161 were measured as 0.43 μM, 1.18 μM, 1.17 μM, 2.00 μM, 4.32 μM, and 4.21 μM for the respective cell lines (Figure 3I). These findings suggest that the presence of the *XIRP2* mutation may confer the heightened resistance of HCC cells to fludarabine and oxaliplatin while simultaneously enhancing their sensitivity to WEHI-539 and LCL 161.

### 3.4. XIRP2 Mutation Increased the Protein Stability of XIRP2 Protein

Additionally, we conducted an investigation into the impact of the *XIRP2* mutation on *XIRP2* gene expression. By analyzing the transcriptional levels of *XIRP2* in HCC tissues from the TCGA and ICGC cohorts, we observed no significant difference in the transcriptional levels of *XIRP2* between HCC tissues with and without the *XIRP2* mutation (Figure 4A,B). Similarly, using qRT-PCR in HCC cells, we discovered no significant difference in the *XIRP2* mRNA expression between the HCC cells harboring the *XIRP2* mutation and those with the *XIRP2* wildtype (Figure 4C). However, in contrast to the mRNA levels, a higher level of XIRP2 protein was observed in the HCC cells harboring the *XIRP2* mutation as compared to those with the *XIRP2* wildtype (Figure 4D). In order to investigate the underlying mechanism, cyclohexane (1 μM; CHX) was utilized to inhibit protein synthesis. It was found that the protein degradation of XIRP2 in HCC cells harboring the *XIRP2* mutation was lower than that in cells with the *XIRP2* wildtype (Figure 4E,F). These findings suggest that the single nucleotide missense mutation of XIRP2 leads to the increased protein stability of coded protein.

### 3.5. Increased XIRP2 Protein in HCC Cells with XIRP2 Wildtype Reduced the Sensitivity to Oxaliplatin

In the clinical setting, oxaliplatin is commonly used as a first-line treatment for patients with HCC. Consequently, our study aimed to investigate the impact of XIRP2 protein on cellular resistance to oxaliplatin. To achieve this, plasmids were introduced into Huh7 and Huh1 cells to augment the expression of the XIRP2 protein (Figure 5A). Prior research has demonstrated that elevated levels of zinc ions can trigger calcium overload, which in turn enhances cellular vulnerability and increases sensitivity to chemotherapy drugs [17,18]. Based on the findings of the BP analysis, which demonstrated an association between the *XIRP2* mutation and metal ion metabolism, specifically zinc ion (Figure 2G), we proceeded to investigate the potential impact of XIRP2 overexpression on zinc ion and its downstream calcium ion. Our results indicate that the overexpression of XIRP2 protein alone did not significantly affect zinc ion and calcium ion levels in cells, as determined by spectrophotometer analysis (Figure 5B,C). However, when the cells were treated with oxaliplatin, we observed a decrease in the zinc ion and calcium ion levels in Huh7 and Huh1 cells with an XIRP2 overexpression (Figure 5B,C). Immunofluorescence experiments conducted on Huh7 cells have provided additional evidence supporting the notion that oxaliplatin induces an increase in zinc and calcium ions within the cells. Nevertheless, it was observed that the overexpression of XIRP2 can counteract this effect (Figure 5D). Moreover, our investigation revealed that the overexpression of XIRP2 protein does not exert any discernible impact on the proliferation of Huh7 and Huh1 cells (Figure 5E) or the formation of cellular spheres (Figure 5F). However, it was observed that the overexpression of XIRP2 protein significantly mitigated the inhibitory effects of oxaliplatin on cell proliferation and sphere formation (Figure 5E,F). Similarly, through performing a TUNEL stain, we found that the overexpression of XIRP2 protein in Huh7 and Huh1 cells did not affect cell apoptosis but significantly reduced the apoptosis induced by oxaliplatin (Figure 5G).

### 3.6. Knockdown of XIRP2 in HCC Cells with the XIRP2 Mutation Increased the Sensitivity to Oxaliplatin via Inducing the Overload of Zinc and Calcium Ion

Subsequently, we investigated the impact of XIRP2 in HCC cells harboring the XIRP2 mutation. To achieve reduced XIRP2 expression in SNU475 cells, two specific siRNAs were employed (Figure 6A,B). Our findings revealed that XIRP2 knockdown in SNU475 cells had no discernible effect on cellular zinc levels (Figure 6C) and calcium levels (Figure 6D), as determined through spectrophotometric analysis. However, upon exposure to oxaliplatin, the levels of zinc ions and calcium ions exhibited a significant increase as compared to the control group (Figure 6C,D). The immunofluorescence findings were in agreement with the spectrophotometer results (Figure 6E). Additionally, our investigation revealed that the reduction of XIRP2 expression in SNU475 cells substantially enhanced the suppressive impact of oxaliplatin on cell proliferation and sphere formation (Figure 6F,G). Moreover, the administration of DP-b99, a chelating agent for zinc and calcium ions, exhibited slight effects on inhibiting cell proliferation but effectively reversed the sensitization caused by XIRP2 knockdown (Figure 6F,G). Furthermore, we found that the knockdown of XIRP2 significantly increased the apoptosis rate in cells with XIRP2 knockdown, and the administration of DP-b99 can partly reverse the effects induced by the XIRP2 knockdown (Figure 6H).

## 4. Discussion

Genomic instability, characterized by various chromosomal alterations including deletions, duplications, rearrangements, and inversions, is commonly observed in the majority of cancer cells. This phenomenon is driven by a multitude of genetic mutations, which subsequently give rise to distinct acquired traits in cancer cells, such as uncontrolled proliferation, the evasion of immune surveillance, tissue infiltration, metastasis, and resistance to therapeutic agents [5]. Notably, HCC exhibits pronounced drug resistance and metastatic potential, resulting in diminished quality of life and elevated mortality rates among affected patients [19]. The genetic alterations in HCC have been extensively characterized in recent years, with notable mutations identified in *CTNNB1* and *Braf* [16]. It was revealed that p53 haploinsufficiency contributes to the invasiveness of HCC cells by enhancing the activation of the PI3K/AKT pathway [20]. Additionally, a study [21] demonstrated that the *BRAF* V600E mutation amplifies the tumorigenic effects of HCC cells. Nevertheless, the precise biological consequences and underlying molecular mechanisms of gene mutations in HCC remain largely unexplored.

In the current study, an examination of the somatic variant profiles of HCC tissues in the TCGA and ICGC was conducted to establish that the *XIRP2* mutation is a prevalent mutation. This mutation was found to be associated with a negative prognosis in HCC. XIRP2 belongs to the actin-binding, xin-repeat-containing protein family and has the potential to maintain the morphology of stereocilia and hearing function [22]. The high frequency of the *XIRP2* mutation has been observed in various diseases, such as high-risk neuroblastoma [23] and breast cancer [24]. However, the investigation of the relationship between the *XIRP2* mutation and patient survival has not been addressed in prior studies. We posit that this study may provide the initial evidence suggesting that the XIRP2 mutation could serve as a prognostic biomarker for HCC patients.

Furthermore, by conducting the DEG analysis and a subsequent enrichment analysis comparing HCC tissues with the *XIRP2* mutation and the *XIRP2* wildtype, it was observed that the *XIRP2* mutation significantly impacts the metal ion metabolism process, specifically zinc ion and copper ion. Metal ions serve as second messengers, swiftly activating numerous signaling pathways upon extracellular stimulation by being released from intracellular stores [25]. Zinc ions, as indispensable metal ions in cellular systems, assume a pivotal function as a fundamental constituent for the biosynthesis of various proteins [26]. Intriguingly, a diminished concentration of zinc has been detected in a range of malignancies, such as prostate cancer [27] and lung cancer [28]. The provision of zinc ion dietary supplements has demonstrated the potential to mitigate tumor development [29]. Prior research has demonstrated that elevated levels of zinc ions can trigger calcium overload, which in turn enhances cellular vulnerability and increases sensitivity to chemotherapy drugs [17,18]. Elevated zinc levels in cancer cells may be a strategy to treat cancers. Consistent with previous studies, via biological experiments, we provided the first evidence that the presence of the *XIRP2* mutation leads to the enhanced stability of the *XIRP2* protein, whereas its depletion significantly heightens sensitivity to oxaliplatin by inducing an excessive influx of zinc and calcium ions.

## 5. Conlusions

In conclusion, our study revealed that the *XIRP2* mutation is prevalent in HCC and is strongly correlated with unfavorable prognosis and drug resistance. The presence of the *XIRP2* mutation leads to the enhanced stability of the *XIRP2* protein, whereas its depletion significantly heightens sensitivity to oxaliplatin by inducing an excessive influx of zinc and calcium ions. Consequently, we propose that the *XIRP2* mutation holds potential as a prognostic biomarker and that targeting the XIRP2 protein may offer therapeutic benefits in the management of HCC.

## Figures and Tables

**Figure 1 biology-13-00633-f001:**
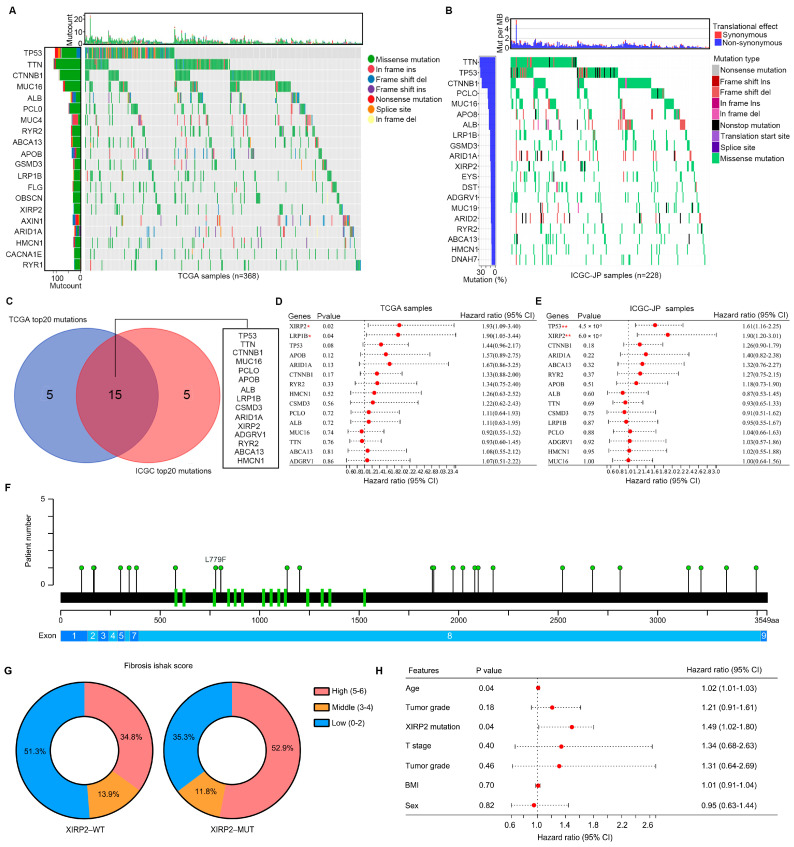
Analysis of mutations in HCC tissues: (**A**) Screening of the top 20 mutation genes from 368 HCC tissues in the TCGA. (**B**) Screening of the top 20 mutation genes from 228 HCC tissues in the ICGC. (**C**) Exploration of the overlapping gene mutations in both the TCGA and ICGC. (**D**) The relationship of the overlapping gene mutations and the overall survival times of HCC patients in the TCGA. (**E**) The relationship of the overlapping gene mutations and the overall survival times of HCC patients in the ICGC. (**F**) Location of the mutation site of the *XIRP2* mutation. aa, amino acid. (**G**) More HCC patients with the *XIRP2* mutation had high fibrosis Ishak scores. (**H**) The *XIRP2* mutation can act as an independent factor for HCC patient survival. * represents *p* < 0.05; ** represents *p* < 0.01.

**Figure 2 biology-13-00633-f002:**
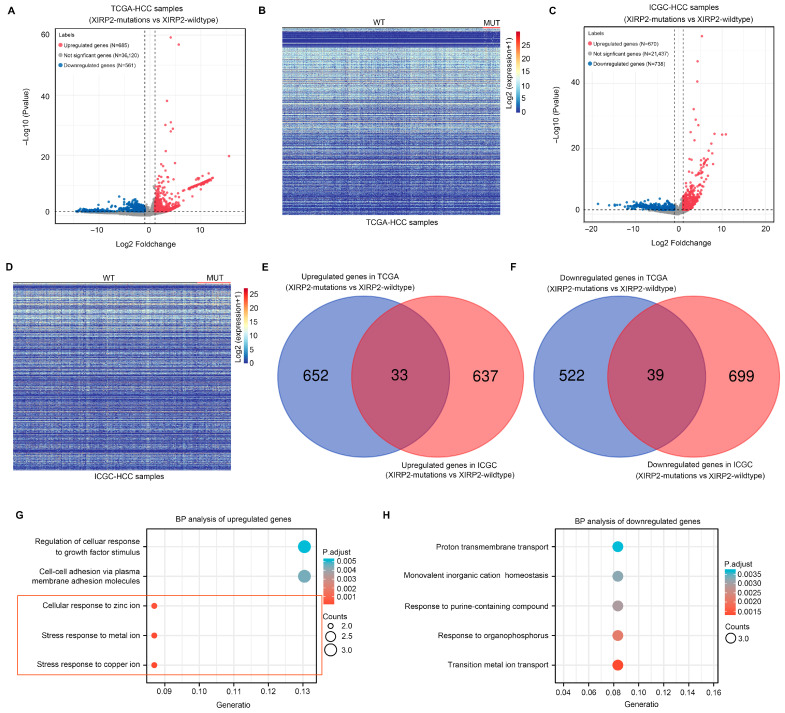
Screening of differentially expressed genes (DEGs) between the *XIRP2* mutation and *XIRP2* wildtype in HCC and the enrichment analysis: (**A**) Volcano plot showing changes of genes between the *XIRP2* mutation and the *XIRP2* wildtype in the TCGA. (**B**) Heatmap showing the DEGs between the *XIRP2* mutation and the *XIRP2* wildtype in the TCGA. (**C**) Volcano plot showing changes of genes between the *XIRP2* mutation and the *XIRP2* wildtype in the ICGC. (**D**) Heatmap showing changes in DEGs between the *XIRP2* mutation and the *XIRP2* wildtype in the ICGC. (**E**) Intersection analysis demonstrated that 33 genes were upregulated in the *XIRP2* mutation and the *XIRP2* wildtype in both the TCGA and ICGC. (**F**) Intersection analysis demonstrated that 39 genes were downregulated in the *XIRP2* mutation and the *XIRP2* wildtype in both the TCGA and ICGC. (**G**) Enrichment analysis of the biological process terms of the 33 overlapping upregulated genes. (**H**) Enrichment analysis of the biological process terms of the 39 overlapping downregulated genes.

**Figure 3 biology-13-00633-f003:**
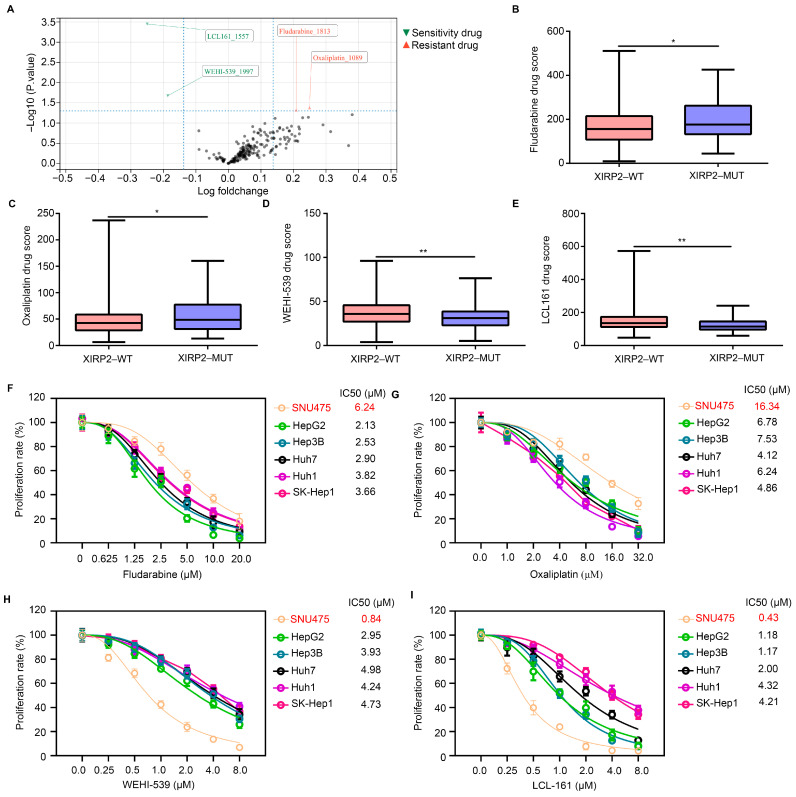
*XIRP2* mutation increased the resistance of HCC cells to fludarabine and oxaliplatin but increased their sensitivity to WEHI-539 and LCL 161. (**A**) The OncoPredict algorithm was used to analyze the change of drug score of 198 drugs between the HCC tissues with the *XIRP2* wildtype and the *XIRP2* mutation. (**B**) Drug score of fludarabine between the HCC tissues with the *XIRP2* wildtype and the *XIRP2* mutation. (**C**) Drug score of oxaliplatin between the HCC tissues with the *XIRP2* wildtype and the *XIRP2* mutation. (**D**) Drug score of WEHI-539 between the HCC tissues with the *XIRP2* wildtype and the *XIRP2* mutation. (**E**) Drug score of LCL-161 between the HCC tissues with the *XIRP2* wildtype and the *XIRP2* mutation. SNU475 harbored the *XIRP2* mutation (I827V), while HepG2, Hep3B, Huh7, Huh1, and SK-Hep1 harbored the *XIRP2* wildtype. (**F**) CCK-8 assays were used to detect the IC50 of fludarabine in SNU475, HepG2, Hep3B, Huh7, Huh1, and SK-Hep1 in 48 h. (**G**) CCK-8 assays was used to detect the IC50 of oxaliplatin in SNU475, HepG2, Hep3B, Huh7, Huh1, and SK-Hep1 in 48 h. (**H**) CCK-8 assays were used to detect the IC50 of WEHI-539 in SNU475, HepG2, Hep3B, Huh7, Huh1, and SK-Hep1 in 48 h. (**I**) CCK-8 assays were used to detect the IC50 of LCL-161 in SNU475, HepG2, Hep3B, Huh7, Huh1, and SK-Hep1 in 48 h. * represents *p* < 0.05; ** represents *p* < 0.01. *n* = 3. The control group was used for comparison. Data are shown as mean ± SD.

**Figure 4 biology-13-00633-f004:**
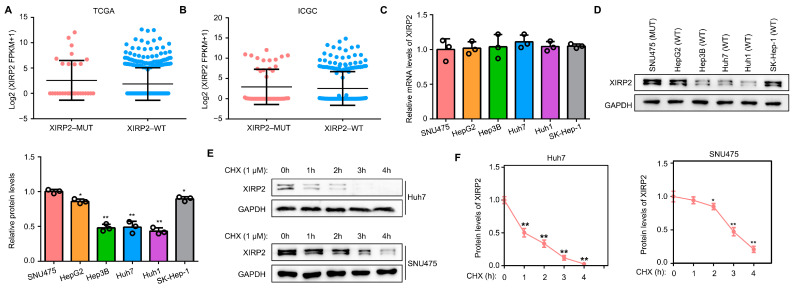
The *XIRP2* mutation increased the protein stability of XIRP2 protein. (**A**) Expression of *XIRP2* mRNA levels in the HCC tissues with the XIRP2 mutation and the XIRP2 wildtype from the TCGA databases. A total of 29 HCC patients with XIRP2-MUT in TCGA cohort (pink dots), while the number of patients with XIRP2-WT was 339 (blue dots). (**B**) Expression of *XIRP2* mRNA levels in HCC tissues with the XIRP2 mutation and the XIRP2 wildtype from the ICGC databases. A total of 21 HCC patients with XIRP2-MUT in ICGC cohort (pink dots), while the number of patients with XIRP2-WT was 238 (blue dots). (**C**) Expression of the *XIRP2* mRNA levels in SNU475, HepG2, Hep3B, Huh7, Huh1, and SK-Hep1. (**D**) Expression of XIRP2 protein levels in SNU475, HepG2, Hep3B, Huh7, Huh1, and SK-Hep1 cells. (**E**,**F**) Degradation rate of XIRP2 protein in SNU475 and Huh7 cells. * represents *p* < 0.05; ** represents *p* < 0.01. *n* = 3. The control group was used for comparison. Data are shown as mean ± SD. Appendix A are Original band for XIRP2 and GAPDH in Figure 4D, Appendix A are Original band for XIRP2 of Huh7, GAPDH of Huh7, XIRP2 of SNU475 and GAPDH of SNU475 in Figure 4E.

**Figure 5 biology-13-00633-f005:**
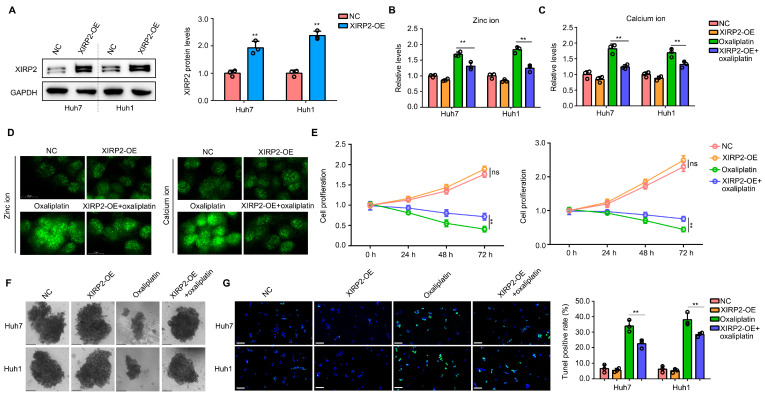
Increased XIRP2 protein in HCC cells with the XIRP2 wildtype reduced their sensitivity to oxaliplatin. (**A**) XIRP2 plasmids were transfected into Huh7 and Huh1 cells to construct XIRP2 overexpression cells. (**B**,**C**) Results of spectrophotometry indicated the levels of zinc ion and calcium ion in Huh7 and Huh1 cells after XIRP2 overexpression and oxaliplatin treatment. (**D**) Results of immunofluorescence indicated the levels of zinc ion and calcium ion in Huh7 cells after XIRP2 overexpression and oxaliplatin treatment. (**E**) Cell proliferation in Huh7 and Huh1 cells after XIRP2 overexpression and oxaliplatin treatment. (**F**) Spherogenesis of Huh7 and Huh1 cells after XIRP2 overexpression and oxaliplatin treatment. (**G**) TUNEL stain of Huh7 and Huh1 cells after XIRP2 overexpression and oxaliplatin treatment. ** represents *p* < 0.01. *n* = 3. The control group was used for comparison. Data are shown as mean ± SD. Appendix A are Original band for XIRP2 and GAPDH in Figure 5A.

**Figure 6 biology-13-00633-f006:**
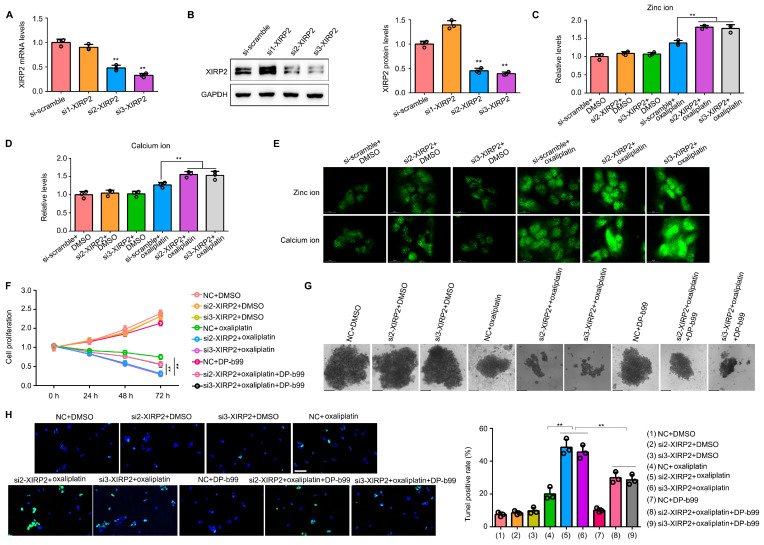
Knockdown of XIRP2 in HCC cells with the XIRP2 mutation increased the sensitivity to oxaliplatin via inducing the overload of zinc and calcium ion. (**A**,**B**) Targeting XIRP2 siRNAs were transfected into the SNU475 cells to construct XIRP2 knockdown cells, and qRT-PCR and western blotting were used to detect the efficiency of the siRNAs. (**C**,**D**) Results of spectrophotometry indicated the levels of zinc ion and calcium ion in the SNU475 cells after XIRP2 knockdown and oxaliplatin treatment. (**E**) Results of immunofluorescence indicated the levels of zinc ion and calcium ion in the SNU475 cells after XIRP2 knockdown and oxaliplatin treatment. (**F**) Cell proliferation in the SNU475 cells after XIRP2 knockdown and oxaliplatin treatment or combined with DP-b99 treatment. (**G**) Spherogenesis of the SNU475 cells after XIRP2 knockdown and oxaliplatin treatment or combined with DP-b99 treatment. (**H**) TUNEL stain of the SNU475 cells after XIRP2 knockdown and oxaliplatin treatment or combined with DP-b99 treatment. ** represents *p* < 0.01. *n* = 3. The control group was used for comparison. Data are shown as mean ± SD. Appendix A are Original band for XIRP2 and GAPDH in Figure 6B.

## Data Availability

Experiment results are available from the corresponding authors.

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
