# Peer review of "Identification of Clinical Value and Biological Effects of XIRP2 Mutation in Hepatocellular Carcinoma"

_biology, 2024, doi:10.3390/biology13080633_

Round 1

Reviewer 1 Report

Comments and Suggestions for Authors

Have your investigators tested patients with cirrhosis for XIRP2 with the hope of earlier recognition of an HCC ? Have you tested HCC patients in China to ascertain if the results can impact on improving the prognosis with medical or surgical therapy?

Below are my extended comments:

What are the main claims of the paper and how significant are they?

The main claims of the authors are that XIRP2 mutation is a potential biomarker for predicting the prognosis and drug sensitivity for HCC as well serving as a therapeutic target to enhance oxaliplatin.

How does the paper stand out from others in its field?

This paper stands out from others because the spectrum of cases is more extensive.

Are the claims novel? If not, which published papers compromise novelty?

The claims are specific and embrace other published papers on this topic.

Are the claims convincing? If not, what further evidence is needed?

The claims are convincing but require prospective studies in other countries and other parts of the world, e.g. UK, Europe, and the United States.

Are there other experiments or work that would strengthen the paper further?

The treatment of patients that fall within the realm of steatohepatitis, both alcoholic and non-alcoholic where cirrhosis and HCC was not initially apparent, should be studied. Is XIRP2 mutation uncovering earlier phases of cirrhosis and HCC with the hope of an opportunity for improved prognosis?

How much would further work improve it, and how difficult would this be?

This depends on data the investigators already have. This paper comes from China where HCC is endemic, and the investigators are experienced with HCC. Nevertheless, this data is unlikely to be readily available.

Would it take a long time?

Time is of the essence, and this will require years.

  Are the claims appropriately discussed in the context of previous
literature?
I would include the opinion of a genealogist which I certainly am not.

Author Response

Have your investigators tested patients with cirrhosis for XIRP2 with the hope of earlier recognition of an HCC ? Have you tested HCC patients in China to ascertain if the results can impact on improving the prognosis with medical or surgical therapy? The claims are convincing but require prospective studies in other countries and other parts of the world, e.g. UK, Europe, and the United States. The treatment of patients that fall within the realm of steatohepatitis, both alcoholic and non-alcoholic where cirrhosis and HCC was not initially apparent, should be studied. Is XIRP2 mutation uncovering earlier phases of cirrhosis and HCC with the hope of an opportunity for improved prognosis? This depends on data the investigators already have. This paper comes from China where HCC is endemic, and the investigators are experienced with HCC. Nevertheless, this data is unlikely to be readily available. Time is of the essence, and this will require years. I would include the opinion of a genealogist which I certainly am not.

Answer: dear reviewer, thank you for your valuable suggestions. We have incorporated your feedback into the revised manuscript and have included additional evidence showing that a higher proportion of HCC patients with XIRP2-mutation exhibit elevated levels of fibrosis Ishak score (Fig. 1G). This suggests a potential association between XIRP2 and genetic mutations in cirrhosis. However, it is important to acknowledge that the strength of this evidence is still limited.  We hope that the publication of this manuscript can serve as a preliminary theoretical foundation for seeking approval for human sample ethics from our hospital or national government. Subsequently, we aim to conduct more extensive research on samples from the Chinese population. We hope to get your understanding.

Reviewer 2 Report

Comments and Suggestions for Authors

The manuscript entitled Identification of clinical value and biological effects of XIRP2 2 mutation in hepatocellular carcinoma by Li et al. is a well-designed study that provides interesting and meaningful results. Some improvements can be made:

1. XIRP2 is overexpressed head and neck cancer and prostate cancer in TCGA database. How prevalent is XIRP2 mutation in HCC patients? Is XIRP2 mutation an independent prognostic marker for HCC?

2. It would help the readers to understand Fig. 4D better by marking the cell lines either WT or the XIRP2 mutated.

3. I suggest the authors replace “Combine” with “XIRP2-OE+Oxaliplatin” in Figure 5 because combine means a combination of two or more compounds, however, in this case, it means that treatment of Oxaliplatin in the XIRP2-overexpressed HCC cells.

4. In the proliferation assay, XIRP2 OE showed minimal although significant prevention of oxaliplatin-induced cytotoxicity, however, in the sphere formation assay, there was a much bigger effect of XIRP2 OE. How do the authors explain this?

5. How do the authors explain in Fig. 6A that HCC cells transfected with siXIRP2_1 showed overexpression of XIRP2? Could it be a technical issue?

6. An NC+DP-b99 control is suggested to be included in the cell proliferation and sphere formation assays to show whether blocking zinc and calcium completely or partially reverses the effects caused by XIRP2.

Author Response

  1. XIRP2 is overexpressed head and neck cancer and prostate cancer in TCGA database. How prevalent is XIRP2 mutation in HCC patients? Is XIRP2 mutation an independent prognostic marker for HCC?

Answer: Dear reviewer, the XIRP2 mutation rates in HCC patients from TCGA cohort and ICGC cohort were 7.8% and 11.0%, especially. Through performing multi-factor regression analysis, we also found that XIRP2 mutation can act as an independent factor for HCC patient survival in TCGA cohort (Fig. 1H). Please check.

  1. It would help the readers to understand Fig. 4D better by marking the cell lines either WT or the XIRP2 mutated.

Answer: Thanks for your suggestions. We marked the cell lines either WT or the XIRP2 mutated in figure 4D. Please check.

  1. I suggest the authors replace “Combine” with “XIRP2-OE+Oxaliplatin” in Figure 5 because combine means a combination of two or more compounds, however, in this case, it means that treatment of Oxaliplatin in the XIRP2-overexpressed HCC cells.

Answer: Thanks for your suggestion. We revised the “combine” as  “XIRP2-OE+Oxaliplatin” in Figure 5. Please check.

  1. In the proliferation assay, XIRP2 OE showed minimal although significant prevention of oxaliplatin-induced cytotoxicity, however, in the sphere formation assay, there was a much bigger effect of XIRP2 OE. How do the authors explain this?

Answer: Dear review, as showed in the materials and methods, CCK8 was performed for 0h-72h, however, the time for the observation of sphere formation assay was 20 days. Therefore,  due to the difference of observation time, the difference between the groups from sphere formation assay will be greater. Moreover, compared to 2D model (CCK8), in 3D sphere, The cells inside the sphere are protected by the cells outside, so the proliferative effects of the cells which were resistant to drugs would be amplified. We hope to get your understanding.

  1. How do the authors explain in Fig. 6A that HCC cells transfected with siXIRP2_1 showed overexpression of XIRP2? Could it be a technical issue?

Answer: Thank you for your advice. Your observations are very careful. In fact, siXIRP2_1 also exhibited very small inhibitory effect on XIRP2 mRNA levels (Figure 6A in revised version). However, going from mRNA to protein also involves the process of translation. In fact, in our previous work, we also found a similar phenomenon, that is, when siRNAs/shRNAs slightly inhibit mRNA expression, cells may compensatively up-regulate the translation rate, etc., resulting in a slight increase in proteins. Therefore, we did not use these inefficient siRNAs/shRNAs in the follow-up experiments, so as to avoid the influence on the subsequent results. We hope to get your understanding.

  1. An NC+DP-b99 control is suggested to be included in the cell proliferation and sphere formation assays to show whether blocking zinc and calcium completely or partially reverses the effects caused by XIRP2.

Answer: Yes, we added. Please checked Fig.6F, 6G and 6H. We found that the administration of DP-b99, a chelating agent for zinc and calcium ions, exhibited slight effects on inhibiting cell prolfieration, but effectively reversed the sensitization caused by XIRP2 knockdown. Please check.

Reviewer 3 Report

Comments and Suggestions for Authors

In this study, the authors investigated the clinical value and biological effects of XIRP2-mutation in HCC. They found that XIRP2 mutation was linked to prognosis and chemotherapeutic responses and increased its protein stability. Thus, the authors concluded that XIRP2 may be a potential biomarker for predicting prognosis and drug sensitivity of HCC. In terms of this manuscript, there are some major comments:

1. All the gene names in the text should be italicized.

2. Why is there such a big difference of the XIRP2 basal levels in Huh7 cells between Fig 4D and Fig 4E at 0h? Did the authors use the different amounts of proteins? If so, I recommend the authors to use the same amounts of proteins at 0h when comparing two groups in Fig 4E.

3. Cell proliferation is an increase in the number of cells as a result of cell division. However, cell count is just the reflection of both cell proliferation and cell apoptosis/death. Therefore, the results from cell count assays cannot reflect cell proliferation. To evaluate the efficacy like shown in Fig 5E and Fig 6E, cell apoptosis assays are warranted.

4. English needs to be carefully edited due to typos and errors in tense and grammar. For example, on Page 11, Line 375, “4×109cells/well” should be “4×109cells/well”, etc.

Comments on the Quality of English Language

Moderate editing of English language required.

Author Response

  1. All the gene names in the text should be italicized.

Answer: Yes, we revised and all the gene names in the text were italicized.

  1. Why is there such a big difference of the XIRP2 basal levels in Huh7 cells between Fig 4D and Fig 4E at 0h? Did the authors use the different amounts of proteins? If so, I recommend the authors to use the same amounts of proteins at 0h when comparing two groups in Fig 4E.

Answer: Dear reviewer, the loading protein was same, but the time of exposure were set as different. In figure 4D, if we set the time of exposure of WB as figure 4E, the bands of huh7 would be same as figure 4E. However, the bands of SNU-475 would be overexposure. Therefore, to avoid the overexposure in the bands, we set the exposure times were different in figure 4D and figure 4E. We hope to get your understanding.

  1. Cell proliferation is an increase in the number of cells as a result of cell division. However, cell count is just the reflection of both cell proliferation and cell apoptosis/death. Therefore, the results from cell count assays cannot reflect cell proliferation. To evaluate the efficacy like shown in Fig 5E and Fig 6E, cell apoptosis assays are warranted.

Answer: Thanks for your suggestion. Tunel stain was performed in figure 5G and figure 6H. Please checked.

  1. English needs to be carefully edited due to typos and errors in tense and grammar. For example, on Page 11, Line 375, “4×109 cells/well” should be “4×109 cells/well”, etc.

Answer: Thanks for your suggestion, we checked and revised some errors. If you consider the manuscript need to be revised by native speaker, please guide further.

Round 2

Reviewer 3 Report

Comments and Suggestions for Authors

After the 1st-round revision, the manuscript improves a lot but still needs some revisions. The answer to the Question 2 in the previous response letter is not acceptable, since the comparison between groups must be based on the data under the same conditions which means to use the same amounts of proteins and the same exposure time, etc. If the authors worried about that the bands of XIRP2 in SNU-475 cells were overexposed, why not to set the same exposure time of Huh7 as SNU-475 instead of using different exposure time by each? Different exposure time between groups sometimes causes a bias in the comparison. On the other hand, the big difference of the XIRP2 basal levels in Huh7 cells between Fig 4D and 4E also creates confusion.

Comments on the Quality of English Language

Moderate editing of English language required.

Author Response

Answer: Thanks for your further suggestion. Yes, using different exposure time would may some confusion for reader. Therefore, we used the same sample and performed again with the same exposure time. Please check figure 4E.